# Example-Driven Model-Based Reinforcement Learning for Solving Long-Horizon Visuomotor Tasks

**Bohan Wu, Suraj Nair, Li Fei-Fei[†], Chelsea Finn[†]**
Stanford University, Stanford, CA
{bohanwu, surajn, feifeili, cbfinn}@cs.stanford.edu

**Abstract:** In this paper, we study the problem of learning a repertoire of low-level skills from raw images that can be sequenced to complete long-horizon visuomotor tasks. Reinforcement learning (RL) is a promising approach for acquiring short-horizon skills autonomously. However, the focus of RL algorithms has largely been on the success of those individual skills, more so than learning and grounding a large repertoire of skills that can be sequenced to complete extended multi-stage tasks. The latter demands robustness and persistence, as errors in skills can compound over time, and may require the robot to have a number of primitive skills in its repertoire, rather than just one. To this end, we introduce EMBR, a model-based RL method for learning primitive skills that are suitable for completing long-horizon visuomotor tasks. EMBR learns and plans using a learned model, critic, and success classifier, where the success classifier serves both as a reward function for RL and as a grounding mechanism to continuously detect if the robot should retry a skill when unsuccessful or under perturbations. Further, the learned model is task-agnostic and trained using data from all skills, enabling the robot to efficiently learn a number of distinct primitives. These visuomotor primitive skills and their associated pre- and post-conditions can then be directly combined with off-the-shelf symbolic planners to complete long-horizon tasks. On a Franka Emika robot arm, we find that EMBR enables the robot to complete three long-horizon visuomotor tasks at 85% success rate, such as organizing a desk, a cabinet, and drawers, which require sequencing up to 12 skills, involve 14 unique learned primitives, and demand generalization to novel objects.

**Keywords:** model-based reinforcement learning, long-horizon tasks

## 1 Introduction

We want robots to robustly complete a variety of long-horizon visuomotor manipulation tasks. Such tasks can be completed by composing a sequence of low-level primitive skills, otherwise known as "manipulation primitives" [1] or "options" [2]. For example, the long-horizon task of "organizing an office desk" often means performing a long sequence of shorter tasks, such as picking objects up, opening or closing drawers, placing objects into drawers, and putting pens into pen-holders. However, completing such a long-horizon task may require the robot to learn many low-level visuomotor skills, compose them sequentially, and generalize across novel objects. During task execution, the robot also needs to detect its own failures (e.g. accidentally dropping an object) and correct them (e.g. picking the object up again) before continuing to the remaining parts of the task. We hope to endow robots with such generalization and robustness.

Approaches such as symbolic planning [3], hierarchical RL [4, 5, 6, 7, 8], and hierarchical planning [9] can accomplish compositional task generalization by reasoning in an abstract or symbolic space. While these approaches have shown promising results on tasks such as simulated tool use [10] and non-vision-based manipulation [9], they rely on the ability to acquire a repertoire of robust, low-level motor skills. These low-level skills can in principle be acquired through reinforcement learning, but a number of challenges remain. First, the acquisition of robust and persistent low-level visuomotor manipulation skills is still challenging, and the errors in completing any sub-sequence of

---

[†]Equal advising and contribution.

5th Conference on Robot Learning (CoRL 2021), London, UK.

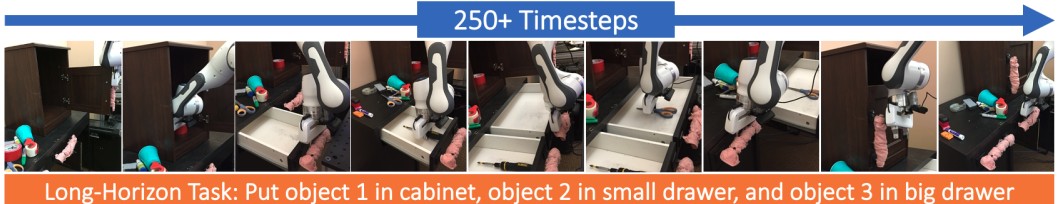

Figure 1: **Example-Driven Model-Based RL (EMBR).** EMBR enables a Franka Emika robot to complete long-horizon manipulation tasks by sequencing up to 12 unique skills at 85% success with novel objects from raw image observations. Here, the robot organizes a desk, which involves a sequence of 10 unique skills.

a long-horizon task can compound over time and lead to overall task failure. Second, acquiring the requisite low-level visuomotor skills requires data-efficient learning and reward or task specification across a broad set of skills. Finally, sequencing the learned primitive skills to perform a long-horizon task requires persistent grounding of the robot's visual observation within the broader abstract plan.

To address these challenges, we introduce example-driven model-based RL (EMBR), an algorithm that learns a dynamics model and a set of Q-functions and success classifiers to acquire a repertoire of visuomotor skills that are both robust and can be directly grounded into symbols for symbolic planning of a long-horizon visuomotor task. More specifically, the success classifiers in EMBR enable both robustness and skill grounding, as they allow the robot to determine the pre- and post-conditions of a skill. For example, a reward function and post-condition for the skill of opening a drawer can be obtained from learning a binary image classifier of "whether the drawer is open" given the robot's pixel observation. These success classifiers require a modest number of example images, making it easy to specify a variety of skills. Finally, by learning a task-agnostic dynamics model, EMBR allows data to be shared across all tasks, leading to greater data-efficiency.

The main contribution of this work is a framework for learning grounded visuomotor skills that can be directly sequenced by off-the-shelf symbolic planners to complete long-horizon tasks. Experiments on a Franka Emika Panda robot arm find that EMBR enables the robot to complete three long-horizon tasks from raw pixels with 85% success, such as organizing a desk, a cabinet, and two drawers, which require up to 14 learned unique primitives, sequencing up to 12 skills, and generalization to novel objects. Moreover, our ablation studies confirm that all components of our method including the dynamics model, success classifiers, Q-functions, and stages are essential to performance for tackling long-horizon tasks.

## 2   Related Work

Reinforcement learning (RL) has a rich history of being used for robotic control problems on tasks from locomotion [11, 12, 13] to object manipulation [14, 15, 16]. In this work, we focus on learning *vision-based* robotic manipulation skills, a problem studied by many prior works [17, 18, 19, 20]. Model-free deep RL has been effective on short-horizon skills like object grasping [21, 20, 22], pushing and throwing [19, 23, 24], and multi-task learning [23, 25] from images. Alternatively, model-based approaches [26, 27], which explicitly learn the environment forward dynamics from images have also been employed for multi-task vision-based robotic manipulation tasks, either with planning algorithms [28, 29, 30, 31, 32, 33, 34, 35] or for optimizing a parametric policy [36]. Unlike these prior works, our algorithm uses a model in conjunction with task-specific success classifiers and Q-functions to learn a wide range of skills suitable for sequencing into long-horizon tasks. In Section 5, we find that both components of our approach are essential for robust performance.

Even with powerful reinforcement learning methods, task specification on real robots remains challenging, as engineering rewards on physical systems can be costly and time-consuming [37]. Motivated by this, many works have studied the reward specification, including inverse reinforcement learning [38] with robot demonstrations [39, 40, 41, 42, 43], learning rewards from user preferences [44, 45, 46, 47], and learning rewards from videos of humans [48, 49]. One common approach in visual RL is to learn to reach a goal image using a coarse measure of reward like negative $\ell_2$ pixel distance [30, 50, 51] or temporal distance [52, 53, 34]. While these techniques have had some success on real robots, such rewards often provide a sparse and difficult to optimize reward signal. Alternatively, a number of recent works learn a classifier from a modest number of examples goal states [54, 55, 24, 56, 57, 25] and tries to learn agents which maximize the classifier score. In this work, we extend this approach to multiple stages of human-provided goals, which we find enables

more effective learning of hard exploration skills, and find that such reward specification can be used to learn 14 unique visuomotor skills on a real robot without any demonstrations.

Like many prior works, the goal of this work is to complete challenging long-horizon visuomotor tasks. One class of prior work is hierarchical RL, which aims to learn temporally-extended primitive skills and a high-level policy over them [4, 5, 6, 7, 8, 58]. Such approaches include jointly learning the primitive skills with the high-level policy through goal generation or an "options" framework [6, 8, 59], learning primitives through intrinsic motivation or other auxiliary objectives [60, 61, 62], from demonstration behavior [63, 64, 65, 66, 67], or from low-level, potentially goal conditioned, reward functions [7, 58, 68, 69]. EMBR is not a hierarchical RL algorithm, as it only learns low-level skills provided to the high-level planner and does not learn a model over options. Nevertheless, our work is similar to the last group, except that we learn visuomotor primitives on a real robot from rewards derived from human examples. Other works have also explored learning hierarchical policies on top of hand-designed skills [70, 71, 72]. These methods are complementary to EMBR, as skills learned via EMBR can be incorporated into most hierarchical policy learning algorithm.

Alternatively, a number of recent works have studied long horizon visual planning [52, 53, 73, 74, 75, 50, 76, 77, 78], often by combining a structured search approach with learned components like generative models [73, 50, 76] or distance functions [53, 34]. While these approaches have demonstrated impressive results on problems such as visual navigation [52, 53], unlike prior work we learn manipulation tasks on a *real robots* that consists of up to 12 unique skills and 250+ timesteps.

Lastly, the task and motion planning (TAMP) literature [79, 80] has extensively studied long-horizon robotic tasks. Unlike these works, we do not assume pre-defined state representations or accurate state estimators, making it possible to handle novel objects in cluttered scenes. Like our work, a number of works have also explored combining learned models and skills with hand-designed or symbolic planning [10, 9, 81] and learning to ground skills for planning [82]. Unlike these techniques, a key insight in this work is that we can jointly learn visuomotor skills and the associated grounding of their pre- and post-conditions, allowing us to leverage off-the-shelf symbolic planners to execute long horizon tasks in a closed-loop fashion directly from pixels on a real robot.

## 3 Preliminaries

**Tasks and skills**. We consider the problem of learning a repertoire of skills that can be sequenced to complete a long-horizon task $\mathcal{M}$. We model the robot's environment as a controlled Markov process $\mathcal{E} = \langle \mathcal{S}, \rho_0, \mathcal{A}, \mathcal{T}, \gamma, H \rangle$, with an image observation space $s \in \mathcal{S}$, an initial state distribution $\rho_0$, an action space $a \in \mathcal{A}$, a dynamics model $\mathcal{T} : \mathcal{S} \times \mathcal{A} \times \mathcal{S} \to \mathbb{R}$, a discount factor $\gamma \in [0, 1)$, and a finite horizon $H$. We notice that many long-horizon tasks are composed of a sequence of lower-level skills. For example, to organize an office desk, the robot needs to acquire a number of skills such as grasping an object, opening a drawer, and placing an object in a drawer. Let $K$ denote the total number of unique skills the robot has acquired, and $k \in [1, K]$ denote the $k^{th}$ skill in the robot's skill repertoire. Depending on the environment state, the order in which the skills should be executed to successfully complete a long-horizon task varies. For example, to organize a desk, the robot may not need to use the skill of closing a drawer if the drawer is already closed. Here, each skill is an MDP $\mathcal{M}^k = \langle \mathcal{E}, \mathcal{R}^k \rangle$, where the robot's environment $\mathcal{E}$ is shared across all tasks and skills, and $\mathcal{R}^k : \mathcal{S} \times \mathcal{A} \to \mathbb{R}$ is the reward function for the $k^{th}$ skill (e.g. the skill of opening a drawer).

**Problem Definition**. Formally, EMBR takes as input $N$ human-provided example images $\Psi^k = s_1, \ldots, s_N$ for each skill $k$ we intend to robot to learn. Our goal (i.e. EMBR's output) is to learn the symbolic grounding of the pre- and post-conditions $g^k_{pre}, g^k_{post} : \mathcal{S} \to \{0, 1\}$, and a policy $\pi^k : \mathcal{S} \to \Pi(\mathcal{A})$ for each skill $k$, where $\Pi(\cdot)$ defines a probability distribution over a set. Each policy has a Q-function, $Q^k : \mathcal{S} \times \mathcal{A} \to \mathbb{R}$ for taking action $a$ from $s$ and following $\pi^k$ onward.

**Variational Autoencoders (VAEs)**. VAEs compress high-dimensional observations $s$ such as images into an embedding $z$. VAEs can be optimized by maximizing the evidence lower bound (ELBO): $\max_{p,q} \mathcal{L}_{\text{vae}}(s)$, where $\mathcal{L}_{\text{vae}}(s) = \mathbb{E}_{q(z|s)} \left[ \log p \left( s \mid z \right) \right] - D_{\text{KL}} \left( q(z \mid s) \parallel p(z) \right)$, where $p$ denotes the generative model, and $q$ denotes the variational distribution.

**Symbolic planning**. A symbolic planner performs task planning in the symbolic domain. Concretely, given a set of candidate actions, each action's associated pre- and post-conditions, the current condition $h$ and a goal condition $g$, a symbolic planner outputs the sequence of actions that allows the robot to reach from $h$ to $g$. The pre-condition of an action is a set of predicates that defines

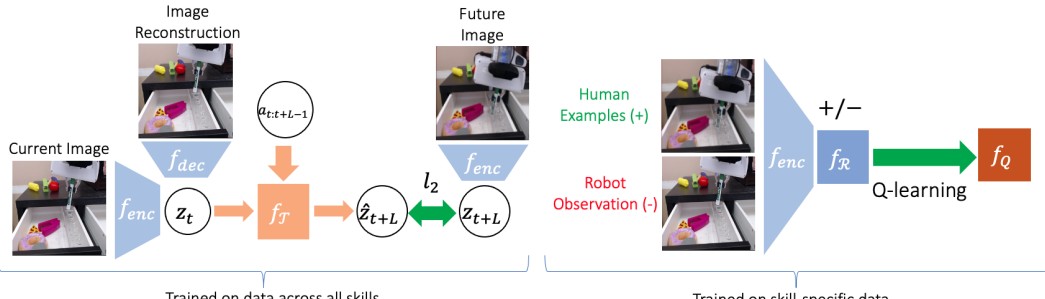

Figure 2: **Example-Driven Model-based RL (EMBR)**. During training, EMBR learns a VAE $f_{vae}$ and a dynamics model $f_{\mathcal{T}}$ in the VAE latent space (**left**). For each skill, EMBR uses human provided example images to train a success-classifier-based reward function $f_{\mathcal{R}}$, and associated Q function $f_Q$ (**right**). At test time actions are selected by model predictive control over the latent dynamics model using the Q-value as the planning cost. The reward function further serves as a success classifier that encourages the robot to reach a successful state, such as inserting the marker as shown in the figure.

the condition that needs to be satisfied in order for the action to be executable. The post-condition of a skill is a set of predicates that defines the effect of executing an action. For example, the pre-condition and post-condition of an "Open left drawer" action is "Drawer is closed" and "Drawer is open" respectively. In our context, the candidate actions are the primitive skills learned by EMBR.

## 4 Example-Driven Model-Based RL (EMBR)

To learn a repertoire of skills that can be directly sequenced by an off-the-shelf symbolic planner to complete long-horizon visuomotor tasks, we propose Example-driven Model-Based RL, or EMBR. Below we first describe how EMBR learns each visuomotor skill without any demonstration skills or trajectories during training, and then elaborate on how these learned skills can be directly sequenced to complete a long-horizon task at test time.

### 4.1 Learning a Repertoire of Visuomotor Skills

During training, EMBR learns a repertoire of low-level persistent visuomotor skills, such as opening a drawer or inserting a marker, using a set of example success images provided by a person for each task. The model learned by EMBR, denoted as $f$, is composed of four components: a global variational autoencoder (VAE) $f_{vae}$, a global latent dynamics model $f_{\mathcal{T}}$, stage-specific binary reward functions $f_{\mathcal{R}}^k$, and stage-specific Q-functions $f_Q^k$. The VAE and latent dynamics model jointly learn a low-dimensional representation of the image observations to address the challenge of efficiently learning many primitive skills; both are trained on data collected across all skills. The skill-specific binary classifiers allow us to learn a reward function for each skill from human-provided images. We use the latent dynamics model to perform model predictive control with the skill-specific Q-function as the terminal planning objective. The skill-specific Q-functions and reward functions are trained on skill-specific data. Next, we elaborate on each of these components.

**Learning a low-dimensional latent space with a VAE and a latent dynamics model**. Performing a long-horizon task using a large number of visuomotor skills demands the algorithm to be sample-efficient and fast at execution time. To address both of these demands, we learn a single VAE $f_{vae} = \{f_{enc}, f_{dec}\}$ to construct a latent space $\mathcal{Z}$. This VAE consists of an encoder $f_{enc} : \mathcal{S} \to \Pi(\mathcal{Z})$ and a decoder $f_{dec} : \mathcal{Z} \to \Pi(\mathcal{S})$. We train $f_{vae}$ using data collected across all skills via Eq. **??**, where $p(s \mid z) \equiv f_{dec}(s_t \mid z_t)$, $p(z) \sim \mathcal{N}(0, 1)$, and $q(z \mid s) \equiv f_{enc}(z_t \mid s_t)$. By learning from all skills, the VAE can improve data efficiency. In practice, we use the greedy hierarchical VAE [51] as $f_{vae}$ with the current image observation as a context frame. Based on this latent space, EMBR learns a latent dynamics model $f_{\mathcal{T}}$ jointly with $f_{vae}$ with data collected across all skills via $l_2$ loss:

$$\min_{f_{vae}, f_{\mathcal{T}}} \mathbb{E}_{s_{t:t+L}, a_{t:t+L} \sim \mathcal{D}} \left[ \sum_{l=1}^{L} -\mathcal{L}_{vae}(s_{t+l-1}) + \left( f_{\mathcal{T}}(z_t, a_{t:t+l-1}) - z_{t+l} \right)^2 \right] \qquad (1)$$

where $z_t \sim f_{enc}(\cdot \mid s_t)$, $L$ is the rollout length of $f_{\mathcal{T}}$ and $\mathcal{D}$ is the dataset collected by the robot.

**Obtaining rewards by learning image classifiers**. Obtaining a reward function in RL typically requires some forms of human supervision such as reward engineering or demonstrations. In EMBR, binary image classifiers serve not only as reward functions for learning primitive visuomotor skills,

but also as success detectors as well as grounding of raw pixels for long-horizon symbolic planning. Concretely, we learn a reward function $f_{\mathcal{R}}^k$ for each skill $\mathcal{M}^k$ by training binary image classifiers to distinguish human-provided example images from robot-collected non-example images using the binary cross-entropy objective [55, 56]. The reward learning objective for skill $k$ is:

$$\max_{f_{\mathcal{R}}^k} \mathbb{E}_{s^+ \sim \mathcal{D}^+, z^+ \sim f_{\text{enc}}(\cdot | s^+)} \left[ \log \left( f_{\mathcal{R}}^k(z^+) \right) \right] + \mathbb{E}_{s^- \sim \mathcal{D}, z^+ \sim f_{\text{enc}}(\cdot | s^+)} \left[ \log \left( 1 - f_{\mathcal{R}}^k(z^-) \right) \right] \quad (2)$$

Here, $\mathcal{D}^+$ is the dataset of images labeled as positive by a person (typically 100-200 images), $s^+$ are the images sampled from $\mathcal{D}^+$, and $s^-$ are sampled from all images in $\mathcal{D}$, the dataset collected by the robot, and are considered negative by default. In our experiments, these positive examples can be collected in 15-30 minutes per visuomotor skill, and 345 minutes in total for all 14 skills. No demonstration trajectories are collected.

At test time, the learned image classifiers are re-purposed as post-condition grounding of each skill for long-horizon task execution. Some of the pre-conditions the task planner requires can be derived from existing post-conditions. For example, for a drawer-opening skill, the pre-condition of "drawer is not open" can be derived from the post-condition of "drawer is open" by negating this post-condition. For the remaining pre-conditions that cannot be derived from any existing post-conditions, we train additional image classifiers using Eq. 2.

**Learning Q-functions for model-based control**. While one can directly use the latent dynamics model for model-based planning, the difficulty of learning an accurate dynamics model for the full horizon of a primitive skill increases with the length of the skill itself. To alleviate this challenge and further accelerate skill learning, we learn a Q-function $f_Q^k$ in the latent space. Concretely, we perform Q-learning via the following objective for skill $k$:

$$\min_{f_Q^k} \mathbb{E}_{s_t, a_t, s_{t+1} \sim \mathcal{D}} \left[ f_Q^k(z_t, a_t) - \left( \overline{f_{\mathcal{R}}^k}(z_{t+1}) + \gamma \overline{f_{\mathcal{R}}^k}(z_{t+1}) \max_{a_{t+1}} f_Q^k(z_{t+1}, a_{t+1}) \right) \right]^2 \quad (3)$$

where $\overline{f_{\mathcal{R}}^k}(z) \equiv \mathbb{1}\{f_{\mathcal{R}}^k(z) > 0.5\}$, $z_t \sim f_{\text{vae}}(\cdot \mid s_t)$, $z_{t+1} \sim f_{\text{vae}}(\cdot \mid s_{t+1})$, and the Q-value target is computed by maximizing over $m_1$ randomly and uniformly sampled actions $a_{t+1}^{1:m_1}$.

---

**Algorithm 1** EMBR at Training Time: Learn a Repertoire of Visuomotor Skills

1: **Input**: Example images per skill; **Output**: $K$ primitive visuomotor skills
2: $\mathcal{D} \leftarrow$ to empty dataset, $f \leftarrow$ random weights, $J \leftarrow$ gradient update steps, $L \leftarrow$ rollout horizon
3: **for** Skill $k \in [1, K]$ **do**
4:     **while** Not reaching target success rate for skill $k$ as determined by success classifier $f_{\mathcal{R}}^k$ **do**
5:         Collect trajectory $d = \{s_{1:H}, a_{1:H}\}$ using actions selected by Eq. 4 and append to dataset $\mathcal{D}$
6:         **for** $j = 1 : J$ **do**
7:             Sample a minibatch of trajectories of timesteps $L \leq H$: $\overline{\mathcal{D}} = \{s_{t:t+L}^{1:B}, a_{t:t+L}^{1:B}\} \sim \mathcal{D}$
8:             Update $f_{\text{vae}}$, $f_{\mathcal{T}}$ jointly via Eq. 1, $f_{\mathcal{R}}^k$ via Eq. 2, and $f_Q^k$ via Eq. 3, all with $\overline{\mathcal{D}}$

---

**Algorithm 2** EMBR at Test Time: Long-Horizon Task Execution via Skill Composition

1: **Input:** trained model $f$, symbolic goal condition $g$
2: Compute current symbolic state $h$ using $f_{\mathcal{R}}$; $s \leftarrow$ new image observation
3: **while** $h \neq g$ **do**
4:     Compute the first skill to perform $k \in [1, K]$ using symbolic planner given goal $g$ and current state $h$
5:     **while** $f_{\mathcal{R}}^k(f_{\text{enc}}(s)) < 0.5$ **do**
6:         Execute robot action $a$ via Eq. 4 given $s$
7:         $s \leftarrow$ current image observation
8:     Compute current symbolic condition $h$ from $f_{\mathcal{R}}$

---

At both training and test time, we perform model-based planning in the latent space by first rolling out $f_{\mathcal{T}}$ for $L$ timesteps into the future and then evaluating the predicted future latent space observation at timestep $t + L$ using the learned Q-function, a process that is illustrated in Fig. 2:

$$a_t = \left( \arg\max_{a_{t:t+T-1}, a_{t+L} \in \mathcal{A}} f_Q^k \left( f_{\mathcal{T}}(z_t, a_{t:t+L-1}), a_{t+L} \right) \right)_1, \text{ where } z_t \sim f_{\text{vae}}(\cdot \mid s_t) \quad (4)$$

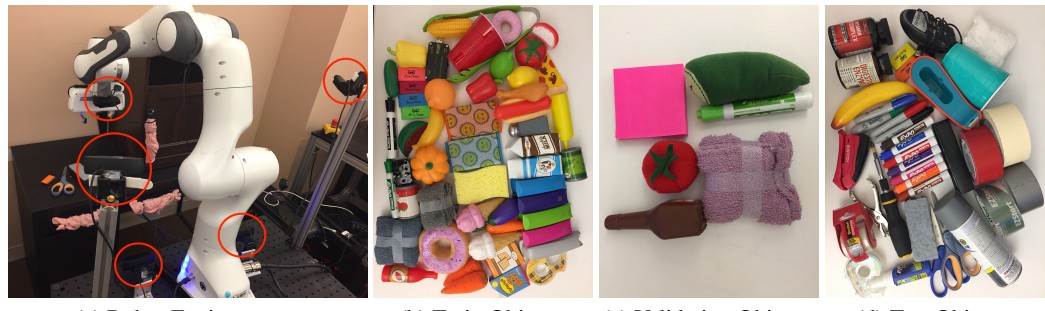

| (a) Robot Environment | (b) Train Objects | (c) Validation Objects | (d) Test Objects |

Figure 3: **Real-Robot Experimental Setup**. In Fig. **(a)**, a Franka Emika Panda robot is equipped with five RGB cameras (red circles) capturing 64×64 RGB images. Quantitative justification for having five cameras is in Appendix F. The robot interacts with a desk that has two drawers and a cabinet, as identified by three pink handles in Fig. **(a)**. Fig. **(b), (c) and (d)** visualize train, validation, and test objects used in all experiments.

Here, the outer and inner maximization operations are taken over $m_0$ and $m_1$ samples respectively.

Integrating these four components, EMBR is trained simultaneously across all components from scratch (i.e. without pre-training) for learning a repertoire of visuomotor primitive skills (Alg. 1). Concretely, the robot begins with an empty dataset $\mathcal{D}$ and can learn each skill in almost any order (Appendix C.2). During skill learning, the robot collects a new trajectory using actions selected from Eq. 4 and updates all model components using $\mathcal{D}$. This process repeats until the current skill reaches a desirable success rate as computed by the success classifiers, before the robot moves on to learn the next skill. As the robot acquires more skills, the dataset $\mathcal{D}$ accumulates and improves the generalization of the visual dynamics model, which in-turn accelerates future skill learning. Wall-clock training time per skill and hyperparameters are detailed in Appendix G and H respectively.

## 4.2 Composing Learned Visuomotor Primitive Skills for Long-Horizon Task Execution

At test time, EMBR provides the learned primitive skills and the symbolic grounding of each skill's post-conditions (in the form of success classifiers) to a high-level task planner (standard PDDL STRIPS symbolic planner [83] in our case) to perform long-horizon task. As elaborated in Alg. 2, the human first specifies a long-horizon task using a symbolic goal condition $g$. Next, EMBR computes the current symbolic condition of the environment from raw pixel observations by passing the current image observations to each of the success classifiers. Using both the symbolic goal condition and the current symbolic condition, the symbolic planner computes the sequence of skills to perform and executes the first skill of this sequence using actions selected from Eq. 4. This procedure repeats until the current condition matches the goal condition, after which task execution terminates. Note that the arm position of the robot is reset after executing each skill. The predicate definitions, the pre- and post-conditions and the full details of the symbolic planner used in Section 5 are in Appendix C.3.

## 5 Experiments

Our experiments aim to answer four key questions: 1) how does EMBR compare to prior RL algorithms on both performing primitive skills and when sequencing those skills to complete long-horizon tasks? 2) how necessary are stage-specific reward functions (as elaborated in Appendix **??**) compared to a single skill-specific reward function? 3) how important is replanning using the classifiers for the robustness of the algorithm? and 4) how important is each of the five camera observations to the performance of the primitive skills? To answer these questions, we conduct experiments across 14 primitive skills and three long-horizon tasks, using the experimental setup in Fig. 3, in which a Franka Emika Panda robot has access to five cameras, each capturing 64×64 RGB images. One of the five cameras is mounted on the wrist. To test object generalization, all objects used in all evaluations are novel and unseen during training, except for the desk, drawers, cabinet, and the marker holder. Fig. 3 visualizes the train, validation, and test objects used in all experiments. See Appendix C for experimental details. Project website at https://sites.google.com/view/embr-site.

**Comparisons.** We compare EMBR to three prior methods: **(1)** EMBR w/o $f_{\mathcal{T}}$, which corresponds to EMBR without the dynamics model, i.e. using only the Q-function and the VAE for action selection; **(2)** EMBR w/o $f_{\text{vae}}$, $f_{\mathcal{T}}$ (Qt-Opt) [20], which is our re-implemented version of Qt-Opt [20] with a

Table 1: Successful trials (out of 20) and success rates of visuomotor primitive skills. Here, EMBR outperforms the three prior methods by 10-20%, 15-25%, and 50-65% respectively on high-precision tasks.

| $k$ | Primitive Skill | Precision | EMBR (Ours) | Prior Method 1: EMBR w/o $f_{\mathcal{T}}$ | Prior Method 2: EMBR w/o $f_{\text{vae}}$, $f_{\mathcal{T}}$ (Qt-Opt) [20] | Prior Method 3: BEE [56] | Ablation 1: EMBR w/ Skill-specific Rewards |
|---|---|---|---|---|---|---|---|
| 1 | Insert marker into a marker holder | High | 20 (100%) | 16 (80%) | 15 (75%) | 8 (40%) | 20 (100%) |
| 2 | Grasp object from left drawer (single) | High | 18 (90%) | 15 (75%) | 14 (70%) | 8 (40%) | 15 (75%) |
|   | Grasp object from left drawer (clutter) | High | 18 (90%) | 14 (70%) | 15 (75%) | 7 (35%) | 14 (70%) |
| 3 | Grasp object from right drawer (single) | High | 18 (90%) | 16 (80%) | 14 (70%) | 5 (25%) | 16 (80%) |
|   | Grasp object from right drawer (clutter) | High | 18 (90%) | 15 (75%) | 15 (75%) | 5 (25%) | 16 (80%) |
| 4 | Grasp object from desk (single) | High | 18 (90%) | 16 (80%) | 15 (75%) | 5 (25%) | 12 (60%) |
|   | Grasp object from desk (clutter) | High | 18 (90%) | 16 (80%) | 15 (75%) | 6 (30%) | 12 (60%) |
| 5 | Open right drawer | Medium | 20 (100%) | 18 (90%) | 16 (80%) | 14 (70%) | 17 (85%) |
| 6 | Open left drawer | Medium | 20 (100%) | 19 (95%) | 16 (80%) | 8 (40%) | 16 (80%) |
| 7 | Open cabinet | Medium | 20 (100%) | 18 (90%) | 17 (85%) | 12 (60%) | 18 (90%) |
| 8 | Place object on desk | Low | 20 (100%) | 20 (100%) | 20 (100%) | 20 (100%) | 20 (100%) |
| 9 | Place object in cabinet | Low | 20 (100%) | 20 (100%) | 20 (100%) | 20 (100%) | 20 (100%) |
| 10 | Place object into left drawer | Low | 20 (100%) | 20 (100%) | 20 (100%) | 20 (100%) | 20 (100%) |
| 11 | Place object into right drawer | Low | 20 (100%) | 20 (100%) | 20 (100%) | 20 (100%) | 20 (100%) |
| 12 | Close right drawer | Low | 20 (100%) | 20 (100%) | 20 (100%) | 20 (100%) | 20 (100%) |
| 13 | Close left drawer | Low | 20 (100%) | 20 (100%) | 20 (100%) | 20 (100%) | 20 (100%) |
| 14 | Close cabinet door | Low | 20 (100%) | 20 (100%) | 20 (100%) | 20 (100%) | 20 (100%) |

Table 2: Successful trials (out of 20) and success rates of long-horizon tasks. $K^*$ and $K$ refer to the minimum number of skills that need to be sequenced and the minimum number of *unique* skills required, respectively. Here, EMBR outperforms the three prior methods by 15-25%, 20-30%, and 85% respectively.

| | Long-horizon Task | $K^*$ | $K$ | EMBR (Ours) | Prior Method 1: EMBR w/o $f_{\mathcal{T}}$ | Prior Method 2: EMBR w/o $f_{\text{vae}}$, $f_{\mathcal{T}}$ (Qt-Opt) [20] | Prior Method 3: BEE [56] | Ablation 2: EMBR w/o Replanning |
|---|---|---|---|---|---|---|---|---|
| 1 | Organize Desk and Cabinet | 8 | 4 | 17 (85%) | 14 (70%) | 11 (55%) | 0 (0%) | 11 (55%) |
| 2 | Organize Markers | 9 | 5 | 17 (85%) | 13 (65%) | 11 (55%) | 0 (0%) | 10 (50%) |
| 3 | Rearrange Objects | 12 | 12 | 17 (85%) | 15 (75%) | 13 (65%) | 0 (0%) | 11 (55%) |

VICE-like success classifier [55] that does not learn a latent space and uses only the Q-function for action selection; and **(3)** BEE [56], which uses the success classifier instead of the Q-function for model predictive control. All comparisons are representative of prior methods for vision-based robotic RL. For all experiments, the data used to train each method is the same dataset collected by EMBR using Alg. 1, which contains 300-400 trajectories collected in 16 hours per skill on average. This provides a more direct comparison and in effect makes the prior methods stronger since the challenge of online exploration is addressed by EMBR during data collection.

## 5.1 Short-horizon Primitive Skill Performance

To answer the first question, we compare all methods across 14 visuomotor skills. In Table 1, we observe that "EMBR (Ours)" can reliably complete all 14 skills with at least 90% success. Compared to "EMBR w/o $f_{\mathcal{T}}$" and "EMBR w/o $f_{\text{vae}}$, $f_{\mathcal{T}}$ (Qt-Opt)", "EMBR (Ours)" improves success rates by 10-20% and 15-25% across high-precision skills, 5-10% and 15-20% for medium-precision skills and 0% for low-precision skills. We hypothesize that EMBR's improvement over "EMBR w/o $f_{\mathcal{T}}$" is due to its ability to alleviate overestimation of Q-values in the presence of limited training trajectories (300-400 per skill), and the addition success rate improvement of EMBR over "EMBR w/o $f_{\text{vae}}$, $f_{\mathcal{T}}$" is due to less overfitting of the success classifiers and Q-functions in the VAE latent space. Finally, EMBR outperforms BEE [56] by 50-65% on high-precision tasks because BEE [56] cannot capture action consequences beyond its planning horizon and the model is not always accurate for the full horizon of a skill.

## 5.2 Long-horizon Task Performance

Next, we compare EMBR to the three prior methods when sequencing the learned visuomotor skills with symbolic planning to complete three challenging long-horizon tasks (Fig. 4):

1. **Organize Desk and Cabinet (8 Skills)** - putting three novel objects cluttered on the desk into the cabinet.

2. **Organize Markers (9 Skills)** - picking up three markers and inserting them into a holder.[2]

3. **Rearrange Objects (12 Skills)** - picking up three objects and placing them to the desk, cabinet or drawers.

---

[2]In this task, we also equip all methods with a scripted skill that uses the edge of the desk as a supporting point to re-orient a marker.

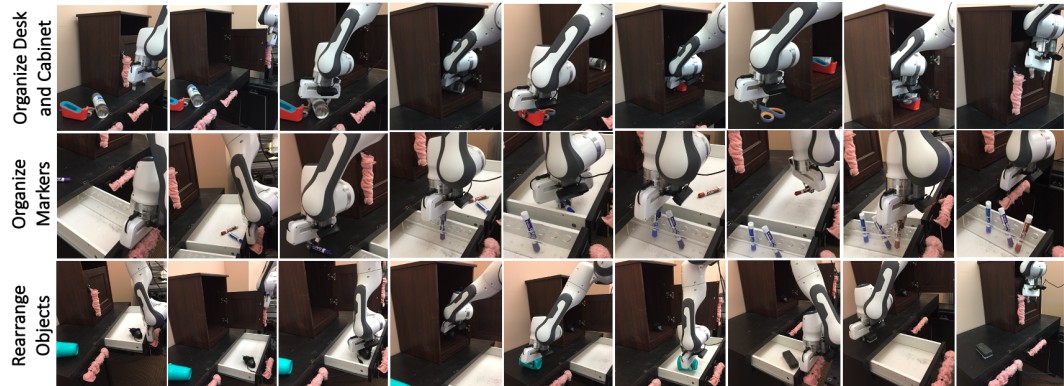

Figure 4: **Long-horizon Task Experiments**. EMBR enables a Franka Emika robot to complete long-horizon manipulation tasks with novel objects from raw image observations. Here, the robot performs three long-horizon tasks with 85% success, which involve a sequence of up to 12 unique primitive visuomotor skills.

Detailed task descriptions are in Appendix C.7. In Table 2, we find that "EMBR (Ours)" completes these long-horizon visuomotor tasks with 85% success, including the most challenging tasks that require up to 12 unique skills. Compared to "EMBR w/o $f_\mathcal{T}$" and "EMBR w/o $f_{\text{vae}}, f_\mathcal{T}$ (Qt-Opt)", EMBR leads to 10-20% and 20-30% improvement in success rates respectively because EMBR achieves higher success rate on individual primitive skills that require higher precision, such as inserting marker and grasping object from desk or drawers. Finally, BEE [56], is unable to perform any long-horizon task due to low performance on skills with medium or high precision. This reflects the importance of learning robust, high-success-rate primitive skills using EMBR.

### 5.3 Ablations

Next, we conduct six ablations to examine the importance of various components of EMBR. Ablation 3, which compares EMBR with vs. without any one camera observation, is in Appendix F. Comparisons to the original BEE and training without stages are in Appendix D and E.

**Ablation 1: Stage-specific vs. skill-specific reward function**. While EMBR accelerates skill acquisition by learning a reward function for each stage of a skill, one can in principle learn just one reward function for each skill. In this ablation, we answer Question 2 by comparing EMBR to an ablated EMBR that learns each skill with a single skill-specific reward function. This does not affect data collection and exploration, which stage-specific rewards can also help. In Table 1, we find that using stage-specific reward functions outperforms using a single skill-specific reward function by 10-30% for higher precision skills such as grasping an object in various locations, indicating the importance of using stage-specific reward functions in long-horizon or higher precision skills.

**Ablation 2: Task performance with vs. without classifier-based replanning**. While EMBR improves the robustness of long-horizon task execution by using success classifiers during planning, one can in principle perform model-based planning without these classifiers. In this ablation, we answer Question 3 by comparing EMBR to an ablation that executes each skill of a task sequentially without using the success classifiers to determine when the skill is successful. In Table 2, we find that long-horizon task execution with classifiers improves performance by 30-35% success rate, due to the robot's ability to detect success in a closed-loop manner until successful. These ablations and Table 2 suggest that *all* components of EMBR are critical to completing long-horizon tasks.

## 6 Conclusion and Future Work

In this work, we propose EMBR: an example-driven model-based reinforcement learning algorithm for learning visuomotor skills that can be directly sequenced to perform long-horizon tasks from raw pixels. Across three long-horizon visuomotor tasks which require up to 14 unique primitives and sequencing up to 12 skills, we find that EMBR consistently achieves 85% success rates, outperforms prior methods, and generalizes to novel objects. Nonetheless, EMBR also has a number of important limitations, including the limited scope of tasks, the amount of supervision required to complete these complex long-horizon tasks, and the limited degree of generalization especially across different scenes. More specifically, the visuomotor skill definitions in EMBR are currently object-specific, such as "open left drawer", which will scale poorly in the presence of many objects. We discuss limitations in more detail in Appendix A, each of which presents interesting directions for future work.

**Acknowledgments**

The authors would like to thank members of the IRIS and RAIL labs for providing valuable feedback. Suraj Nair is funded in part by an NSF GRFP. This work was also supported in part by ONR grants N00014-20-1-2675 and N00014-21-1-2685.

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
