# OpenReview forum: "Example-Driven Model-Based Reinforcement Learning for Solving Long-Horizon Visuomotor Tasks"
_robot-learning.org/CoRL/2021/Conference — CoRL2021 Poster_

### Official Review · Reviewer_REyR · 2021-07-17

**Originality:** Very Good
**Technical Quality:** Excellent
**Clarity Of Presentation:** Excellent
**Impact:** 3

**Recommendation:**

Strong Accept: I recommend accepting the paper and will argue for my recommendation even if other reviewers hold a different opinion.

**Summary:**

The paper presents a method to sequence learned visuomotor skills by learning a representation space with a VAE, a global dynamics model (shared across skills and trained jointly with VAE) and skill-specific reward classifiers and Q-functions. Model-predictive control is used with the Q-functions to execute skills at runtime. The reward classifiers are used as success detectors for when an executed skill is achieved. The Q-functions are learned to account for model inaccuracy and compounding errors over the planning horizon. Long-horizon planning (sequencing skills) is achieved using the skill success classifiers to ground to predicates that can be utilized in a symbolic planner to achieve a particular goal predicate. The predicates are manually defined (in appendix). Experiments are performed on a real robot with long-horizon tasks of cleaning and organizing environments with containers, drawers, cabinets, and various objects.


**Issues:**

EMBER doesn’t make sense as an acronym for Example-driven Model-based RL. It's a real stretch, just to get an acronym that is a nice word. I advise against this, but these acronyms are rampant in deep RL so it'll fit right in.

The description of the symbolic planning in the main paper is lacking in details necessary to understand this aspect of the approach. I would suggest adding a more concrete description of the planner utilized. Symbolic planning is described in Section 4.2 and Algorithm 2, but it is described only at a high-level that a symbolic planner is used. There are details in the tables in the appendix, but it is left unclear what is the symbolic planning algorithm used in the paper. The grounding to a symbolic language suitable for planning is contrived as each skill will have one success classifier, but that classifier will signify potentially a complex combination of predicates (e.g. skill 1 in Table 7 will have one success classifier that binds to a combination of three predicates, symbolic action 1 in Table 6). It will in general be non-trivial for a human to exhaustively consider the logical predicates that are to be satisfied for a particular skill pre- and post-condition. The long-horizon performance depends critically on the ability of the engineers to define these correctly. To the authors' credit, this limitation is mentioned in Appendix H. I only want to highlight the limitation since the long-horizon capabilities of the approach is one of the main novelties, and the method will be susceptible to failure on more complex environments and tasks.

Five cameras is a relatively high number of cameras compared to what is utilized on most research robot platforms. The ablation of the wrist camera is nice, I would also be curious to know how crucial it is to have so many cameras, and if some camera placements are more useful than others. Further ablations on the number and placement of cameras could be informative in extensions to this work. I wouldn’t expect more experiments for the revision, but it would be nice to discuss this in the text.

Some citations that are relevant to include:
    - [1] does model-predictive control using a learned latent dynamics predictor and learned reward predictor (which can be effectively a classifier as a sparse reward).
    - [2] extends [1] to predict Q-values instead of just rewards. However, these works do not try to do long-horizon sequencing and they have not been demonstrated to work on real robot systems.
    - [3] is an info-theoretic MPC formulation that learns Q-functions from real data to mitigate the influence of the planning horizon choice and model-error. I think these papers are sufficiently relevant to the present paper to be cited.

[1] Hafner, Danijar, et al. "Learning latent dynamics for planning from pixels." International Conference on Machine Learning. PMLR, 2019.
[2] Hafner, Danijar, et al. "Dream to Control: Learning Behaviors by Latent Imagination." International Conference on Learning Representations. 2019.
[3] Bhardwaj, Mohak, et al. "Information theoretic model predictive q-learning." Learning for Dynamics and Control. PMLR, 2020.


**Reviewer Expertise:**

Excellent: Expert knowledge on the topic of the paper

**Strengths And Weaknesses:**

Strengths:
    + The method is technically sound and intuitive.
    + The paper is well-written and organized, I especially liked the appendix organization and thoroughness
    + The experiments are thorough and very nicely organized. The precision column Table 1 is fantastic, it is an important attribute to consider in the tasks and makes the reported statistics more interpretable across the different baseline methods.
    + Several relevant baselines were compared to, and there were different experiments for comparing performance of the individual skills.
    + The real-robot results are impressive.
Weaknesses:
    - The connection to symbolic planning is tenuous (see Issues below for further explanation).
    - A number of relevant citations are missing (again see Issues)
    - The method itself is not drastically novel, but this is mitigated by the impressive experimental real-robot results.


**Summary Of Recommendation:**

The method itself is not drastically novel, I find the high-level idea to be a combination of ideas from citations [1-3] provided in the Issues section below. However, demonstrating the efficacy of a visuomotor MBRL algorithm on a real robot system is a major contribution, as it is all too common for experiments to be performed on simple simulated environments. Demonstrating success on long-horizon tasks is even better. A number of relevant baselines were compared and the experiments were insightful, particularly the correlation of skill precision to method and baseline performance. The paper was extremely well-written and organized, and extensive details are provided in the appendix. I think this method provides a good basis for further extensions and research.

---

### Official Review · Reviewer_vnfa · 2021-07-23

**Originality:** Fair
**Technical Quality:** Fair
**Clarity Of Presentation:** Good
**Impact:** 3

**Recommendation:**

Weak Reject: I recommend rejecting the paper, but will not argue for my recommendation if the majority of other reviewers have a different opinion.

**Summary:**

This paper proposes a reinforcement learning architecture that can be supplemented by a classical symbolic planner to solve so-called long-horizon tasks. These tasks require sequencing different 'skills' to realize a given goal (a set of logical states). These skills are grounded action-object-location tuples (e.g., 'open left-drawer', 'grasp marker from left-drawer') and comprise either one or two stages. The proposed learning approach relies on stage-specific components (Q-functions, rewards, binary classifier). The results illustrate the success of the algorithm and how it compares against several other methods.

**Issues:**

In addition to my suggestions above (Strengths & Weaknesses), here I provide some further improvement points.

- Skill vs. stage distinction seems rather redundant/arbitrary. Why, for example, 'place object on desk' is a different skill than 'place object in cabinet' or 'open left drawer' vs. 'open right drawer'? Isn't 'place' (or 'open', 'grasp', 'close') a skill by itself? Otherwise, we need to iterate over all possible actions and objects and create a new 'skill', which results in a combinatorial explosion of number of skills that robots will not ever be able to learn.

- The camera setup is unrealistic. The need for 5 cameras just for ultimately pick-and-place tasks have not been discussed or justified (except the importance of the wrist camera, which is appreciated).

- The paper requires going back and forth between the main text and the supplementary material a lot to better understand and evaluate the experimental and implementation details, along with arguments and discussions that are of critical importance for the work.

- The compared algorithms are not exact replicas (different architectures, input spaces, etc.) and above else they are targeted for different scenarios (i.e., short horizon tasks), I'm not sure then whether these are fair comparisons.

- Human collected images do not seem to be always accurate or representative as shown in Fig.5 in the supp. material, or is it due to the low resolution that it's harder to understand?

- Robot policy execution speed is very slow, why is that so?

**Reviewer Expertise:**

Very good: Comprehensive knowledge of the area

**Strengths And Weaknesses:**

The main strength of the paper is its attempt at solving multi-step tasks via a combination of reinforcement learning and classical planning formulation, compared to the focus on learning low-level manipulation skills within RL research domain.
Both the training and testing happen on the physical robot, which also enhances the value proposition of the paper.


Weaknesses:\
The critical weakness of this work is its non-scalable skill definition. The distinction between a skill and a stage seems somewhat unnecessary. Skills as they are defined in this work sound more like a command, and/or impose a hierarchy of skills (stages as they are called in this work). Wording choice might be unimportant, but this choice affects the analysis and comparison to different methods. For example, a comparison to a hierarchical RL framework, or a model-based task and motion planner might be more relevant when terminology and what they imply are better/properly framed. Because as the authors also highlighted classical RL methods are not designed for multi-step tasks.

Second, the learned system is strongly biased on the specific environment, i.e., the robot actually does not learn how to open a drawer, but rather it learns how to open that specific (left/right) drawer. A reliable and effective visuomotor skill learning system should not depend on the left/rightness of a drawer at all.

Third significant weakness is the reliance on a large set of perception systems (here five cameras). This hinders the applicability of this framework in many realistic settings. Furthermore, there isn't any analysis and/or discussion on this design choice, except for the inclusion of the wrist camera.

**Summary Of Recommendation:**

This work is a good initial step towards getting closer to what model-based task and motion planners can already reliably accomplish, which is to solve multi-step reasoning and manipulation tasks. It effectively uses visual perception to ground logical predicates, which are then used for a binary classifier and reward function. However, the skill definition is not scalable, and the learned policies are too dependent on the actual setup which in turn impedes the generalization of the skills to different environments. The visual perception setup, comprising five cameras, is also demanding, hindering the applicability, transfer, and replicability of the framework.

---

> ### Comment · Reviewer_REyR · 2021-08-24
> **Regarding scalability of skill definition**
>
> I do not think that the non-scalability of the skill definition should be counted as a weakness of the method, given the authors are utilizing the defacto standard of skill-based planning PDDL which is utilized by most contemporary TAMP methods today. Further, the primitive skills defined in Table 2 are intuitive basic manipulations. It is true those skills can be decomposed further (e.g. elemental grasp, place, push), but what one considers a primitive skill is a design decision, and that will largely determine how "long" the long-horizon planning can be. If Reviewer vnfa wants to maintain the critical weakness of this paper is the skill definition is not scalable, I think the burden is on them to cite a more scalable definition, and describe how the novelty of the present paper is impacted in view of that alternative.

---

### Official Review · Reviewer_Au3T · 2021-07-24

**Originality:** Poor
**Technical Quality:** Poor
**Clarity Of Presentation:** Good
**Impact:** 3

**Recommendation:**

Strong Accept: I recommend accepting the paper and will argue for my recommendation even if other reviewers hold a different opinion.

**Summary:**

The paper presents a skill learning algorithm to solve long horizon tasks, where human teachers provide skill annotation using trajectories and images of the terminal states. These demonstration skills are then used to learn skill termination conditions and skill policies. The skill termination conditions are used to plan with a symbolic planner, where when a skill is chosen to solve a long horizon task, its policy is used to get to the skill termination state. The skill learning framework EMBER uses all skill demonstrations to learn skill policies at the same time, while also learning an underlying latent representation of the skill under use. The paper then presents ablation results and comparisons to some previous methods to show the EMBER's superior performance.



**Issues:**

The paper in its current state seems to overclaim its capacity to learn a model over options. This model's associated skills are defined by the expert as clarified in the appendix. The paper demonstrates a method to then learn the skill policies and terminations which is an easier problem than learning a model.
Moreover a clear section talking about inputs and outputs of the algorithms along with a clear problem definition would help the paper's readability.

After author response:
I have taken the author's response in mind and their improved clarity of presentation of their approach. I move to accept the paper, and have changed my review accordingly. I also think that the comparison to skill chaining is unnecessary and not an apples to apples comparison.

**Reviewer Expertise:**

Excellent: Expert knowledge on the topic of the paper

**Strengths And Weaknesses:**

Strength:
The paper uses latent representations to learn a large number of skills within the same network.
Weaknesses:
Human and expert annotation of skills. Annotations of skills with termination symbols, and overall model over options are hard to specify for for TAMP problems, and even harder to learn from demonstrations. In my opinion the paper solves the easier problem to learn multiple policies using latent variables, but does show how the skills themselves would be specified. Further, the symbolic planning is performed over the sequence of sub-tasks provided by humans and it is not required to be over the skill termination classifiers learned. Moreover, the comparisons are not fair as there are significant changes in the BEE and Qt-Opt comparisons as listed in the appendix. A fair comparison for skill discovery would be skill learning work such as Bagaria et al.

Bagaria et al: Option Discovery using Deep Skill Chaining, Akhil Bagaria, George Konidaris

**Summary Of Recommendation:**

The paper needs to clarify its claims as to whether it is learning a skill model or using human annotations. This is a hard problem, especially to solve using deep methods. A modification of the claims of the paper would be useful.

---

### Meta-Review · Area_Chair_aFdn · 2021-08-06

**Recommendation:** Accept (Poster)
**Confidence:** 4

**Metareview:**

The paper proposes an RL architecture that uses a symbolic planner for tasks that involve many steps.

Quoting the summary of reviewer REyR:
The authors present a sound approach to doing visuomotor skill learning and show very impressive results on a real robot platform.
The experiments are thorough, detailed, and insightful.

The submission contained major weaknesses, in particular, the claims of the paper were incorrectly stated and that skills are supervised by human experts. The authors have addressed these concerns and put a lot of effort into the revision process.
The remaining weakness of scalability is discussed in the appendix. I ask the authors to move some of this discussion to the main text.

I recommend accepting this paper.

---

> ### Public Comment · ~Andrew_Hundt1 · 2021-09-24
> **Relevant prior work with long horizon multi-step tasks and a real robot**
>
> Overall, this paper looks pretty awesome! These are very challenging tasks, and it has a neat approach. I just saw a couple of small items which all concern Related Work:
>
> 1. lines 77 and 108: The "unlike prior work" statements could use revision in light of [1], which completes raw-image vison-based long-horizon manipulation tasks on a real robot with a Q-function. However, important differences remain in favor of this submission, so I'd expect rewording it to be straightforward. [2] also makes a couple of sim-only improvements on long-horizon tasks.
> 2. The exact topic being discussed in each sentence/paragraph of the Related Work is a bit confusing.
> 3. Typo line 105: "long horizon" should be "long-horizon" (missing hyphen)
> 4. Typo line 109: "on a real robots" should be "on a real robot"
>
> ---
>
> [1] A. Hundt et al., "“Good Robot!”: Efficient Reinforcement Learning for Multi-Step Visual Tasks with Sim to Real Transfer," in IEEE Robotics and Automation Letters, vol. 5, no. 4, pp. 6724-6731, Oct. 2020, doi: https://doi.org/10.1109/LRA.2020.3015448.
>
> [2] Sulabh Kumra, Shirin Josh, Ferat Sahin "Learning Robotic Manipulation Tasks through Visual Planning" ArXiV e-prints https://arxiv.org/abs/2103.01434

---

> > ### Author Response · Authors · 2021-09-27
> > **Response to Public Comments**
> >
> > Hi Andrew, thanks for the comments, and here are our responses to each concern you raised:
> > 1. This is a great point, and we will add discussions of these two references into the paper's related work section.
> >
> > 2. Each paragraph in the related works discusses a sub-category of related methods: robotic RL from images, task specification, hierarchical RL, long-horizon visual planning, and TAMP. We will try to make the organization of the related work section more clear.
> >
> > 3. Many thanks for catching the typos - we will update the paper accordingly. Same response to comment 4.
> >
> > Thank you.

---

### Decision · Program_Chairs · 2021-09-13

**Decision:**

Accept (Poster)

**Comment:**

The paper proposes an RL architecture that uses a symbolic planner for tasks that involve many steps.

Quoting the summary of reviewer REyR:
The authors present a sound approach to doing visuomotor skill learning and show very impressive results on a real robot platform.
The experiments are thorough, detailed, and insightful.

The submission contained major weaknesses, in particular, the claims of the paper were incorrectly stated and that skills are supervised by human experts. The authors have addressed these concerns and put a lot of effort into the revision process.
The remaining weakness of scalability is discussed in the appendix. I ask the authors to move some of this discussion to the main text.

I recommend accepting this paper.